# A Research Platform for Autonomous Vehicles Technologies Research in the Insurance Sector

**Miguel Ángel de Miguel [1], Francisco Miguel Moreno [1], Pablo Marín-Plaza [1], Abdulla Al-Kaff [1], Martín Palos [1], David Martín [1], Rodrigo Encinar-Martín [2] and Fernando García [1,\*]**

[1] Intelligent Systems Lab (LSI) Research Group, Universidad Carlos III de Madrid (UC3M), 28911 Leganés, Spain; mimiguel@ing.uc3m.es (M.Á.d.M.); franmore@ing.uc3m.es (F.M.M.); pamarinp@ing.uc3m.es (P.M.-P.); akaff@ing.uc3m.es (A.A.-K.); mpalos@ing.uc3m.es (M.P.); dmgomez@ing.uc3m.es (D.M.)

[2] Departamento de Movilidad, CESVIMAP-MAPFRE, 05004 Ávila, Spain; rencin1@cesvimap.com

\* Correspondence: fegarcia@ing.uc3m.es

**Abstract:** This work presents a novel platform for autonomous vehicle technologies research for the insurance sector. The platform has been collaboratively developed by the insurance company MAPFRE-CESVIMAP, Universidad Carlos III de Madrid and INSIA of the Universidad Politécnica de Madrid. The high-level architecture and several autonomous vehicle technologies developed using the framework of this collaboration are introduced and described in this work. Computer vision technologies for environment perception, V2X communication capabilities, enhanced localization, human–machine interaction and self awareness are among the technologies which have been developed and tested. Some use cases that validate the technologies presented in the platform are also presented; these use cases include public demonstrations, tests of the technologies and international competitions for self-driving technologies.

**Keywords:** autonomous vehicle; self driving; intelligent architecture; insurance

## 1. Introduction

Autonomous vehicle technologies have received increasing attention in recent years because of their numerous applications and the important economic and social progress that these types of systems represent.

On one hand, from the social point of view, numerous articles and studies have shown the potential progress represented by these technologies; on the other hand, from the industrial point of view, many companies and research centers have sought to perfect these systems. Numerous projects and demonstrations have been presented that have endeavored to solve the challenges that these technologies are still facing today [1]. The insurance sector is one of the different economic sectors that will be affected by the advance of autonomous driving technologies. In this sense, the lack of standardized models has made it very difficult for insurance companies to identify and foresee the costs associated with this new paradigm. More efficient and safer sensors will lead to the reduction of costs associated with accidents and victims; however, the costs associated with the installation and repair of these sophisticated devices will be considerably increased. Furthermore, the new interaction paradigms and liabilities associated with the amortization of road transport open a new method of interpretation of the driving environment and consequently of accidents that occur on the road. All these questions have given rise to an important legal and ethical debate on autonomous driving—debates which insurance companies have not ignored. Furthermore, the technological debate remains open regarding which of the available technologies are safer, which perception systems are more reliable or which means of

communication are more efficient at communicating with other road users. Answers to these and other questions are sought by the new research platform developed by MAPFRE-CESVIMAP, the Institute of Automotive Research INSIA of the Polytechnic University of Madrid and the Laboratory of Intelligent Systems of the University Carlos III of Madrid. The intelligent vehicle of ATLAS (Autonomous Testing Platform for Insurance Research; see Figure 1) has emerged as a research platform for autonomous vehicle technologies, with the aim of testing and proving the advances that have emerged in this regard both from the point of view of high-level control, advanced perception and human factors associated with autonomous driving.

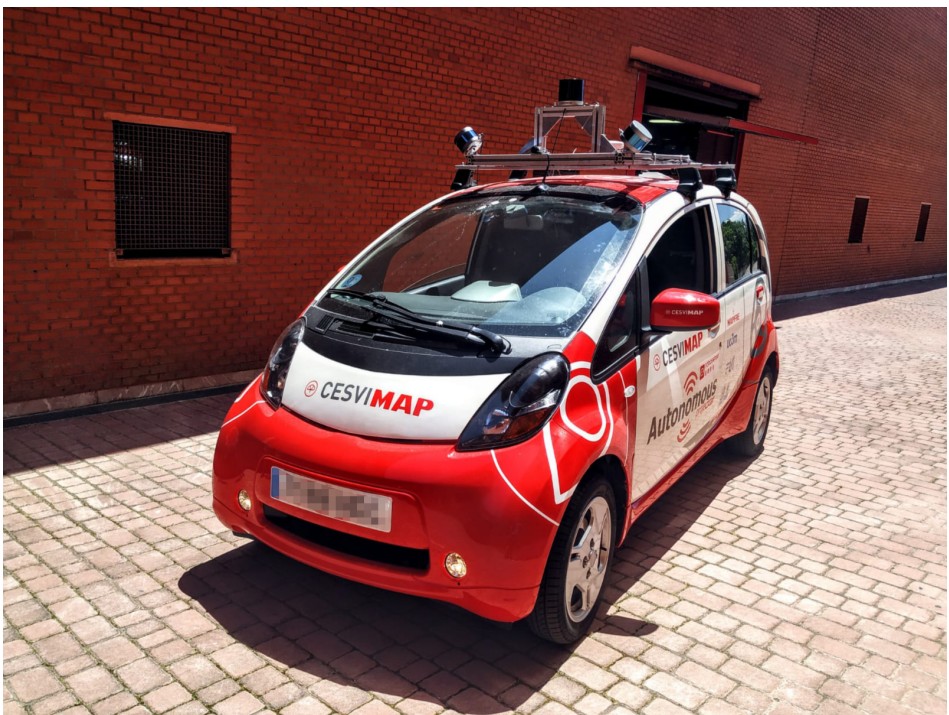

**Figure 1.** Autonomous vehicle platform.

Several examples of autonomous technologies can be found in the literature, and their industrial development is currently being explored by companies such as Cruise, Otto, Uber, Zoox and Waymo. At the same time, some manufacturers are exploring their own independent proposals; for example, the projects of BMW [2] and Mercedes-Benz Bertha [3]. However, to the best of our knowledge, this is the first platform that is dedicated to the research of technologies for the insurance sector.

Autonomous driving will potentially alter one of the main actors in mobility—the insurance sector. The concept of insurance will have to become quite different when based on several new—as yet unknown—indicators. MAPFRE, as a world-wide insurance company, is very focused on new technologies and how these technologies can exert an influence in the future. CESVIMAP, the research center belonging to MAPFRE, has decided to develop its own autonomous platform with LSI-UC3M and INSIA to research how this technology will change the landscape for insurance companies.

This work is organized as follows: First, Section 2 describes the hardware configuration of the vehicle, detailing each module. Afterward, Section 3 describes the software architecture and some of the most relevant algorithms used. Finally, in Section 4 several use cases are presented and, Section 5 exposes the conclusions and future work.

## 2. Hardware Architecture

This section describes the vehicle platform and, more specifically, the hardware elements equipped in the vehicle for high-level perception and control research. The vehicle is designed to become the road version of the first autonomous vehicle of the Intelligent Systems Lab (LSI) [4] iCab. Therefore, this platform

has complex requirements, allowing it to be able to drive in complex environments. The vehicle platform chosen is a Mitsubishi iMiev, which is an electric vehicle that can be driven on roads at up to 130 km/h and that has an autonomy of 160 km. The high level architecture presented in this paper is based on open-source technology; this makes it applicable to different vehicular platforms equipped with drive-by-wire technology. As stated above, the aim of the platform is to foster the testing and development of high-level autonomous vehicles technologies; i.e., high-level control, perception and interaction with the environment. With this intention, the vehicle is equipped with different sensors, computers and communication devices that allow it to drive autonomously, which are detailed in this section.

*2.1. Actuators*

Vehicle drive-by-wire technology was generally introduced in [5] and was tested in the INSIA institute of the Universidad Politecnica de Madrid. After some mechanical and electrical modifications, the vehicle used in this work was made fully drive-by-wire capable. The modifications performed allowed full control via a CAN bus; i.e., steering wheel, brake control and vehicle controls can be totally overridden by the human driver, allowing safe maneuvering and testing.

*2.2. Sensors*

The sensor setup of an autonomous vehicle is, perhaps, the most critical aspect of the design. This is mainly due to the significant cost and to the fact that sensors are essential for autonomous driving. The configuration must include the minimum number of sensors to make sure that all elements surrounding the vehicle are correctly perceived, that it is completely secure and that it follows all driving rules. A review of different sensors and configurations for perception tasks in autonomous driving can be found in [1]. Based on previous experience, and after analyzing the behavior of different sensor setups in a simulated environment, the most adequate set of sensors that provides sufficient information about the environment consists of the following:

**LiDAR**: The proposed sensor setup includes three light detection and ranging (LiDAR) sensors. One 32-layer LiDAR is used as the main sensor, and two 16-layer sensors are used to cover blind spots and to complement the information provided by the main LiDAR. The purpose of these sensors is twofold: on the one hand, the accuracy of the distance estimation of LiDAR sensors is remarkable for localization purposes, as detailed in Section 3.3; on the other hand, these sensors are also used for obstacle detection and localization, directly providing the 3D pose of the different objects in the scene. The selection of this set provides full 360° coverage without compromising the computational load.

**Monocular camera**: Three monocular cameras are equipped and installed on the roof of the car, with an optical horizontal field of view of 90°. This camera covers most of the front area of the vehicle and provides color images that are used later for obstacle classification (e.g., pedestrian, car, bike, etc.) and to obtain an insightful analysis of the road environment, including road lane topology, traffic signs, etc.

**RTK GNSS and IMU**: In order to obtain the precise localization of the vehicle, a GNSS with RTK corrections, fused with an inertial measurement unit (IMU) sensor, is used based on Pix Hawk technology. This system generates precise coordinates and the orientation of the vehicle.

**Vehicle sensors**: In addition to the extra sensors installed, the data available in the CAN bus from the vehicle sensors are also integrated, such as the wheel odometers and the steering angle sensors.

Besides the sensor selection, it is also of great importance to determine the physical location of each sensor on the vehicle. Since this is a research platform, it is common to modify the position of the sensors, as well as to add new ones to perform experiments. For this reason, we have installed a metal rack on top of the roof where sensors can be easily installed using screws. On this rack, the monocular camera is positioned in the center of the front bar providing 189 degrees vision, so they have no occlusions from any other sensor, or from other parts of the vehicle. Then, the main LiDAR is installed at the center, with a small structure that provides an extra elevation from the roof and

the rack. Regarding the two complementary 16 layer LiDARs, they are positioned at the left/right of the other LiDAR, but with a lower elevation and an extra inclination angle that allows them to cover the blind areas of the main LiDAR. Finally, the GNSS and IMU units are positioned at the back of the rack. Last, but not least, in order to combine and fuse the information received from each sensor, we need to know the relative position and orientation between them with a very high accuracy. Using the algorithm presented in [6], it is possible to perform an automatic calibration of these extrinsic parameters for LiDAR and camera sensors with a high accuracy.

*2.3. Processing Units*

Every sensor produces a high amount of data that needs to be processed, and processing units can also be an expensive part of an autonomous vehicle. For this reason, the minimum amount of CPU and GPU power required to run all the algorithms in real time must be determined. In the proposed platform, we divide the computing between three different computers, allowing us to execute multiple processes in parallel, thus distributing and equalizing the computing load. Each processing unit has different capabilities, which have been selected specifically for the tasks to be executed.

**Control computer:** This computer manages the communication with the car systems via the CAN bus. Additionally, it is responsible for executing the control and path planning algorithms that allow the vehicle to move autonomously. It is based on a fast CPU that is capable of executing the required operations in real-time.

**Localization and mapping computer:** This computer is responsible for processing the huge amount of data provided by the three LiDARs and generating the LiDAR odometry and mapping. Furthermore, it also performs all other localization tasks involving the GNSS and the IMU sensors. Since most of the LiDAR processing algorithms are only implemented by the CPU and are not available with GPU acceleration, this computer is only equipped with a fast processor and 16 GB of RAM memory, which is sufficient to process all data at the sensor rate.

**Perception computer**: The perception computer is responsible for executing all the deep learning algorithms used for obstacle classification and scene understanding. Thus, the computing capabilities of this computer are more specific, and this computer is therefore equipped with a high-end NVIDIA Titan-XP GPU with 12 GB of graphic memory, which allows the vehicle to execute all the required algorithms. Finally, this computer is also connected to a screen inside the car, which acts as an interface with the user.

*2.4. Communication*

All the raw data provided by the sensors, and the processed data generated by the computers, need to be shared. Data from one sensor might be needed by more than one computer, and the processed data generated by an algorithm from one computer may be needed by another computer. In order to share all this information, a star topology is adopted in which all the devices are connected to a gigabit switch. The main advantage of this topology is that it is easily extensible if more sensors or computers are added to the network.

Apart from the communications between the systems inside the vehicle network, an Internet connection is also required in order to obtain GNSS differential corrections, download digital maps or connect to V2X networks. For that purpose, a 4G router is connected to the switch, providing Internet access to every computer in the network.

**3. Software Architecture**

The presented platform is governed by a custom software architecture. Although other architecture implementations for autonomous driving exist, and even though they could be suitable for this application, the creation of our own design provides an extra level of flexibility, which is helpful

for research. This is specially useful when designing very specific experiments, as it allows a more customizable integration of the software modules required for the experiment.

The proposed software architecture is based on the Robotic Operating System (ROS). ROS is a framework for developing and integrating robotics applications and is also widely used in autonomous driving software. It provides an inter-process communication middleware and allows the deployment of different software modules (nodes) in a distributed way. In addition, ROS provides a set of tools to aid the researcher in developing and debugging tasks.

### 3.1. Architecture Overview

Figure 2 shows the main modules that constitute the proposed software architecture for the vehicle. Each software module is responsible for executing a specific task, and the connections between modules follow predefined formats based on ROS interfaces. This allows the interchangeability of different components that solve the same task, as long as they follow the specified input and output interfaces. Thus, the modularity present in the architecture is ideal for comparing and testing different solutions, making this architecture unique in terms of research.

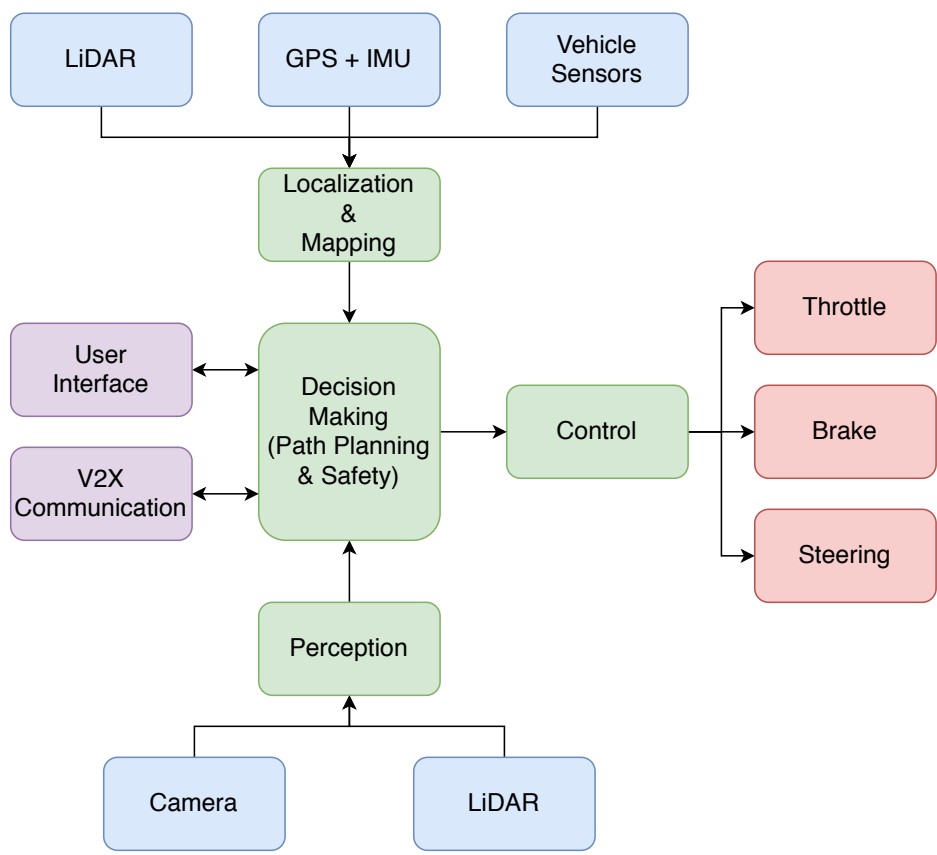

**Figure 2.** Proposed software architecture.

With this layout, data flow from the sensors to the multiple processing modules, ultimately generating the driving commands for steering, acceleration and braking. The vehicle is able to communicate with the users through a graphical user interface, on which the destination goal can be specified. Furthermore, communication with other entities is also achieved through a V2X module.

The next subsections present the modules of the proposed architecture, describing the different algorithms used and how the data flow is handled.

### 3.2. Acquisition

The data acquisition modules are mainly composed of the several drivers that read the raw data from the sensors. Once the raw data are received, the main tasks of these modules are twofold: on the one hand, the input raw data from the sensors may require the application of a preprocessing step. This preprocessing step may consist of a filtering algorithm to clean the noise from the sensor or another type of algorithm to generate more complex information from the received simple data. On the other hand, these drivers must convert the available sensor data to a ROS message format in order to share it with the rest of the modules in the system. By doing this, the information received from the sensors is converted to standard formats, and so the rest of the components of the architecture become sensor-agnostic and are able to perform their tasks, independently of the sensor used.

### 3.3. Localization

Localization is a critical task for an autonomous vehicle. In order to navigate safely, the vehicle must be localized accurately and with robustness. For this reason, the proposed architecture exploits different sensors and algorithms to perform this task, which can be divided into two types: local and global localization.

The local localization task, which is commonly known as vehicle odometry, computes the movement of the vehicle with respect to its origin position by measuring its translation and rotation. These methods are typically very accurate locally but may suffer from drift, so they are unsuitable for long-term localization. In the proposed platform, two different odometry methods are used:

1. Wheel odometry: This method makes use of the velocity and the steering angle provided by the vehicle. By integrating these measurements, the wheel odometry algorithm is able to provide the translation and rotation of the vehicle.
2. LiDAR odometry: In addition to the wheel odometry method, the proposed approach also includes an exteroceptive odometry method based on LiDAR 3D point clouds. The LiDAR odometry method integrated in the platform is LOAM [7], which is able to produce very reliable odometry in real time while also generating a 3D map of the scenario.

Although the proposed odometry methods have a fast update rate and a high local accuracy, they are insufficient for an autonomous vehicle. On the one hand, local methods suffer from drift because error is accumulated. On the other hand, the localization provided by local methods is referenced to the point at which the vehicle started to move, so they do not provide localization coordinates referenced to a global common frame or a digital map. This global reference is required to navigate in the real world. For this reason, two different global localization methods are also integrated, with the aim of providing the vehicle position with a global reference.

1. GNSS localization: GNSS sensors are currently the most common global localization sources in both autonomous and human-driven cars. This information is also combined with the data provided by the IMU in order to also estimate the orientation of the vehicle. The advantage of these sensors is that the received data do not require any significant processing, other than converting the latitude and longitude coordinates to UTM. However, the precision of the obtained position is often reduced by multi-path propagation effects caused by buildings, or even a loss of signal due to areas such as tunnels or urban canyons. For this reason, although GNSS-based localization provides global positioning, it is not sufficient alone to provide the robust and accurate positioning that an autonomous vehicle requires.
2. AMCL: The adaptive Monte Carlo localization (AMCL) algorithm [8] is capable of computing the global pose of the vehicle inside a reference map. This method, which is based on a particle filter, requires three different inputs: the global map of the area, the current LiDAR measurement and the local odometry of the vehicle. The localization process consists of updating the particle weights based on the cost of matching the LiDAR measurement with the map at the pose given by

each particle. Local odometry is used to propagate the particles in the prediction step. The final estimation is computed from the posterior represented by the particle set.

Each of the different methods proposed for local and global localization has its own benefits and drawbacks. On the one hand, odometry methods are able to provide high local accuracy but suffer from drift, and the obtained pose is not referenced to a global frame. On the other hand, global localization methods are typically inferior in terms of precision and may suffer from jumps and inconsistent estimations. In this architecture, we propose a fusion algorithm based on the H infinite filter [9] to combine the multiple localization sources available in the vehicle, thus exploiting their strengths and compensating their weaknesses. Other alternatives, such as the unscented Kalman filter (UKF), are also viable for the fusion of the different localization outputs.

### 3.4. Mapping

Mapping is the process of modeling the vehicle surroundings, allowing the vehicle to interact with the environment. The maps are used by the vehicle to generate routes and to navigate through the environment. Mapping techniques can be categorized into global and local mapping.

In order to plan and generate routes, a global map of the driving area is required. This map is also employed as a reference for global localization, as stated in the localization section. The global map of an area is usually generated offline by integrating multiple measurements from previous (manually-driven) sequences around the area to be mapped. Some high efficiency SLAM methods such as LOAM [7] offer the possibility of creating accurate maps in real time, although these are subject to drift errors. Additionally, the global map can also be obtained from digital map providers such as OpenStreetMaps [10] as shown in Figure 3.

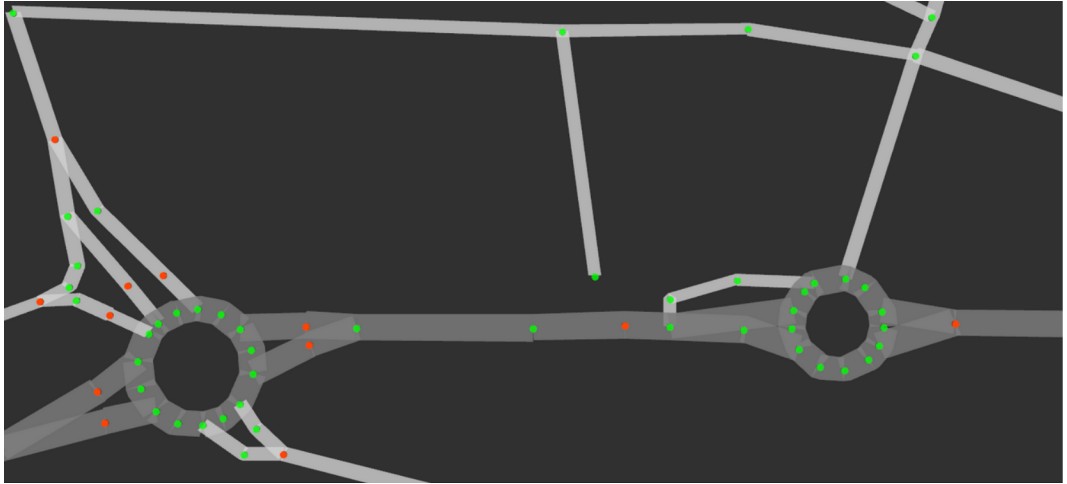

**Figure 3.** Digital map built from OpenStreetMap data.

The main disadvantage of global maps is that they are static. Therefore, although they are used for global route planning, they do not contemplate dynamic obstacles, such as other vehicles or pedestrians, which makes them insufficient for autonomous navigation. In order to compensate for this absence of information, local maps are included in the system. Local maps are generated online and only cover the nearest surroundings of the vehicle, integrating the latest readings from the sensors. In the presented platform, the information from the three LiDAR sensors is processed in real time using the approach presented in [11]. This method generates an occupancy grid map (OGM), which is used later on by the planning and navigation modules in order to handle the existing obstacles around the vehicle. An example of the LiDAR point cloud and the local map output is presented in Figure 4.

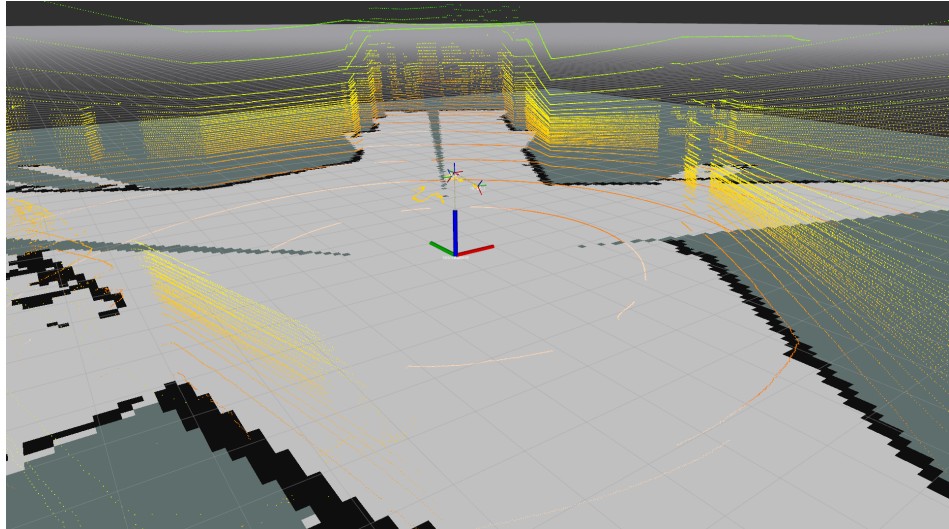

**Figure 4.** Local occupancy grid map generation from light detection and ranging (LiDAR) data.

### 3.5. Perception

The environment perception modules integrated in the vehicle are responsible for analyzing the raw sensor data and extracting meaningful information about the scene. These modules can be categorized into two groups.

- **Obstacle detection and classification:** First, an autonomous vehicle must be able to identify other road agents; i.e., vehicles, pedestrians, cyclists, etc. This difficult task can be performed using different techniques; however, the trend in recent years has been to use deep learning methods to detect and classify the different obstacles around the vehicle. For example, in the presented platform, two different methods are integrated: the first method [12] performs an analysis of the images taken from the monocular camera—this algorithm can reason simultaneously about the location of objects in the image and their orientations on the ground plane—while the second method [13,14] exploits the available 3D information from the LiDAR sensor to also detect and classify the road agents, this time providing 3D detections, as shown in Figure 5. Both methods are able to run in less than 100 ms, thus providing this critical output in real time.
- **Scene understanding:** In addition to obstacles, in order to navigate through the city, an autonomous vehicle must be able to recognize the different elements in the scene. For this reason, several scene understanding methods have also been integrated into the platform. The main algorithm, which is presented in [15] and shown in Figure 6, performs a complete semantic segmentation of camera images, providing information about the different elements in the environment. Additionally, the work in [16] extends this information by adding a deeper semantic analysis of the road layout. Current works also focus on adapting the aforementioned approaches to multiple weather conditions.

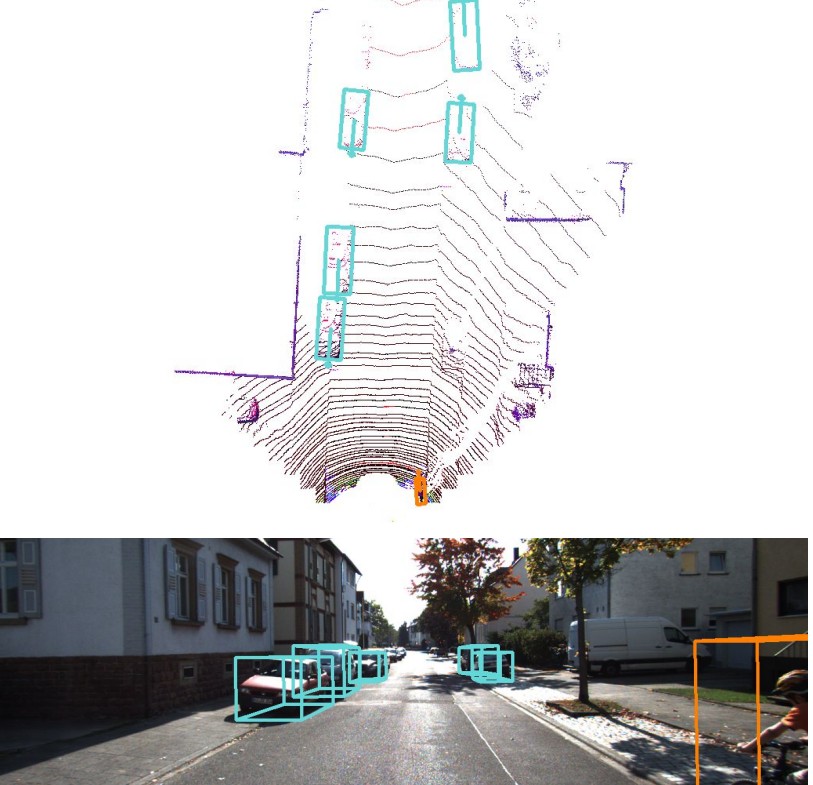

**Figure 5.** LiDAR object detection and classification. Top, bird's eye view of LiDAR points and detections; bottom, their corresponding 3D bounding boxes.

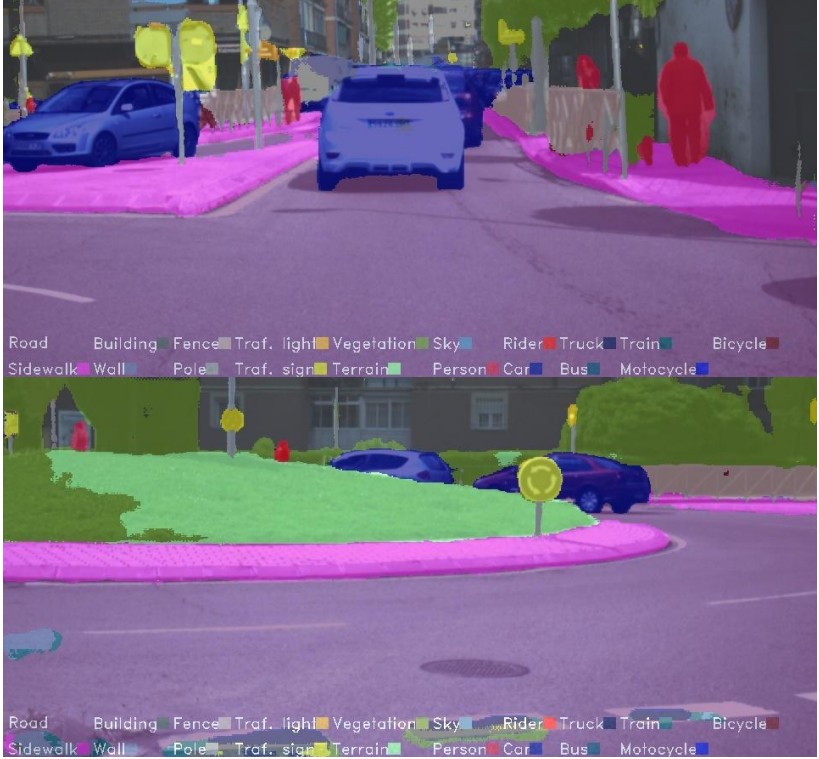

**Figure 6.** Semantic segmentation of a driving scene.

*3.6. Safety*

One of the most critical issues in autonomous vehicles is safety. The vehicle must be able to detect any anomaly or problem and then hand over the driving controls to the human driver. Accordingly, the proposed architecture includes a continuous introspective analysis of the system to evaluate the safety conditions that determine when the vehicle is able to drive in autonomous mode. These conditions are evaluated based on the following factors:

- **Self awareness**: This indicator comes from the self-awareness component, which checks the health of the sensor inputs and the heartbeat of each software process running in the system.
- **Localization accuracy**: If the vehicle is not able to localize itself in the world with sufficient accuracy for autonomous navigation, then the control commands may result in unexpected and undesired behavior. Therefore, this indicator represents the health of the self-localization system, making the human driver take command of the vehicle if necessary.
- **Path unreachable**: This indicator comes from path planning and navigation modules. The human driver may also be required to take control of the car if the vehicle is unable to generate a path to the destination, or in cases where the generated path cannot be followed; e.g., the path goes out of the free navigable space.

*3.7. Path Planning*

The path planning task is in charge of generating a feasible trajectory that the vehicle can follow to reach a goal location, such as in [17,18]. To handle the complexity of generating that trajectory, path planning is usually divided into global planning and local planning.

- Global planning considers all the fixed obstacles, such as the features of the roads (number of lanes, width, connections between different roads, etc.), to find the shortest path between the vehicle's current position and the goal destination.This task is done automatically by obtaining all necessary information from OpenStreetMaps (OSM) and generating the shortest path using Dijkstra's algorithm. The global route also includes information about the maximum speed for each road or the lane in which the vehicle should be [10]. Figure 7 shows a global path (red line) between the current position (blue dot) and the goal destination (green dot), including all the OSM nodes (red dots).
- On the other hand, local planning tries to follow the global plan, except that it considers local obstacles that are not on the global map but are detected with the perception algorithms described above (Figure 8). Besides, the trajectories generated by this module must satisfy some constraints that will make the path followable by the vehicle. All this information is transformed into the frenet frame in order to determine the optimal trajectory, and that trajectory is then transformed back into real-world coordinates so that the control module is able to follow it [19]. Thus, the vehicle is able to overtake slower vehicles if possible or to accelerate or brake accordingly, always maintaining the reference global path and the target speed as goals.

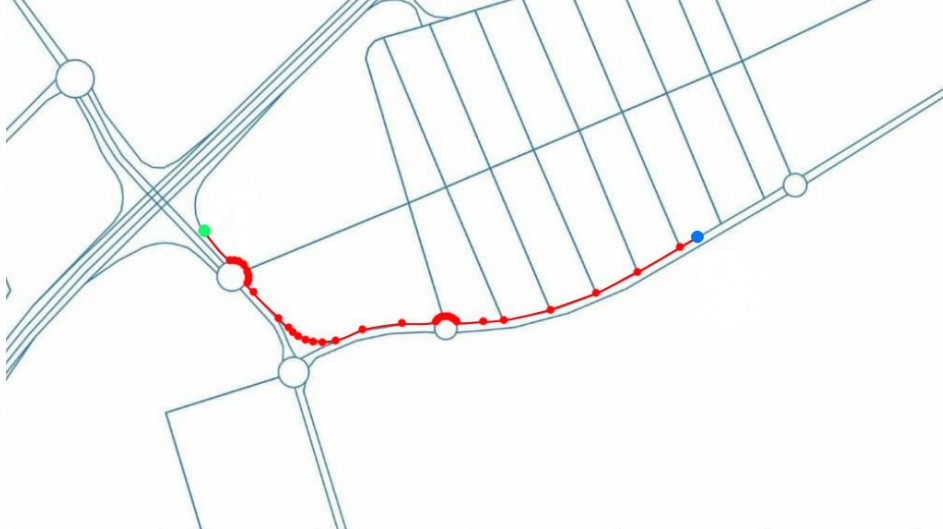

**Figure 7.** Global planning in OpenStreetMaps (OSM).

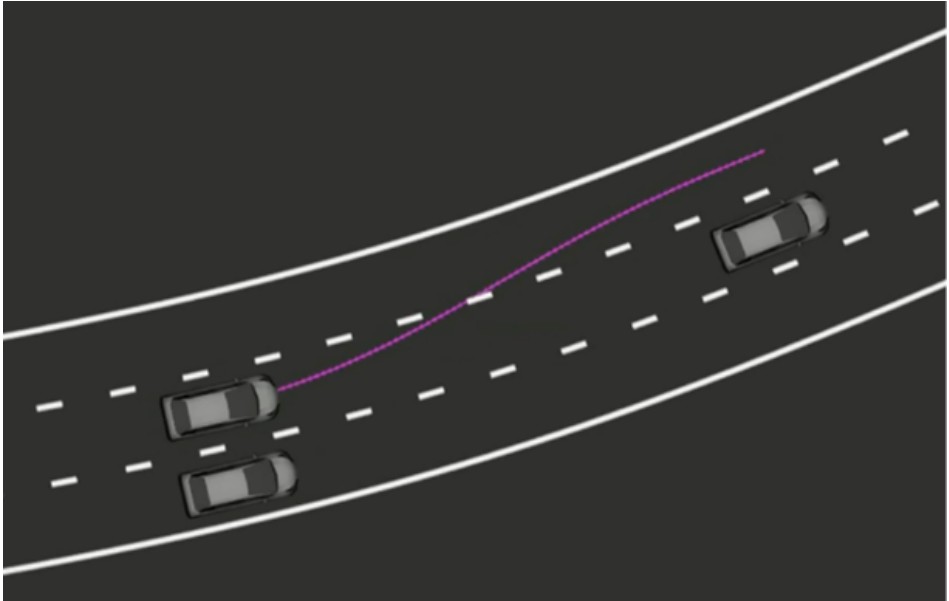

**Figure 8.** Local planning in simulation.

### *3.8. Navigation and Control*

Navigation through the road environment relies greatly on the localization and path planning modules. Once the vehicle is correctly localized, it needs to follow the generated local plan. The path follower algorithm used for this purpose is Stanley [20], which is a lateral controller that outputs the steering angle at each time step, making the vehicle follow the path. The steering commands, together with the target speed provided by the local planner, are sent to the low-level control module that accordingly moves the steering wheel and the throttle and brake pedals [5].

### *3.9. Communication*

In addition to the numerous modules and algorithms presented above, the proposed software architecture also includes a vehicle-to-everything (V2X) communications architecture, based on [21]. Although this is less critical for autonomous navigation, this module provides multiple capabilities which are useful for continuing the ongoing research in smart cities and the cooperation between autonomous vehicles and other entities. A vehicle-to-pedestrian (V2P) application is presented in [22],

where a mobile application is developed to communicate with the vehicle in order to warn pedestrians of nearby vehicles.

In conjunction with the V2X communication architecture, the vehicle can also be equipped with several human–machine Interfaces (HMIs). These interfaces include a large screen to communicate with pedestrians and other road users, and a pedestrian traffic light that is used to indicate if it is safe to cross the road [23].

### 3.10. Simulation

Simulators are increasingly being used in the development of autonomous vehicles as a result of their capability to ease data acquisition and for the design and testing of software features.

In the proposed platform, the CARLA simulator is used to create a digital twin of the real platform inside a simulated environment to achieve fast prototyping and testing, with the latter being vital as, in Spain, it is not legal to test an autonomous vehicle on ordinary roads or under normal traffic conditions.

The CARLA simulator is an open-source autonomous car simulator specialized in urban environments [24]. It is designed on top of Unreal Engine 4—a cutting-edge video game engine—which allows visual information and graphics, physics and environments to be as realistic as the technology allows. In addition, CARLA supports all the common autonomous vehicle sensors with a high level of customization, allowing the replication and simulation of any real sensor, such as those described in Section 2, if parametrized correctly. Furthermore, CARLA is fully compatible with ROS thanks to an official and supported CARLA–ROS bridge, making it an appropriate choice for the proposed architecture.

The complete platform's software architecture described above, except for the communication module, is replicated inside CARLA using ROS as a bridge, which acts as an interface. As the proposed platform's sensor information is presented in a standardized ROS message format, it is possible to substitute the sensor drivers with the artificial sensor data generated by the simulator, consequently allowing the digital twin to perceive the simulated environments as if they were real. Consequently, the cameras, LiDARs, the GNSS and the IMU have been mimicked inside the simulator using their datasheets and CARLA's parametrization feature to achieve the maximum correlation between reality and simulation. Once added, their simulated readings are automatically published through ROS, making them readable by all the platform's software architecture modules. Likewise, the control commands output by the Stanley controller can be directly interfaced with the simulated car through ROS, closing the information loop and transferring the proposed architecture to the simulated environment. In consequence, it is possible to extract information from the digital environment with the several simulated sensors and to drive and act upon the simulated car within the real proposed software architecture.

It should be stressed that no software module is modified in this process; thus, the simulated platform is identical to the physical one, ensuring maximum reliability and correlation and maintaining the crucial platform's modularity and flexibility.

## 4. Use Cases

This section aims to provide a brief demonstration of some of the use cases with the proposed research platform. Note that, because of its high versatility, only some applications are presented, while the actual range of possible uses is actually unlimited.

### 4.1. Multi-Sensor Localization in Urban Environments

Considering localization as one of the most basic needs of an autonomous vehicle, it is of great importance to analyze and compare the multiple methods that are commonly used to solve this problem. The wide selection of sensors equipped in the vehicle makes it a great platform for research in localization techniques. In addition to local odometry methods, such as wheel or LiDAR

odometries, the vehicle can also localize itself globally (in the world and/or inside a map) by making use of the GNSS data and other global localization methods such as Monte Carlo localization (AMCL).

In [8], a method which includes GNSS information in the AMCL algorithm is proposed, which allows the improvement of the distribution of particles in the filter thanks to the GNSS measurements. Furthermore, this method also uses the local odometry generated from the LiDAR sensor to accurately propagate the particles. Figure 9 shows how the LiDAR point cloud (orange) and the map feature (black) matching improves the GNSS localization (green arrows) and makes it converge into a more accurate localization (red arrows).

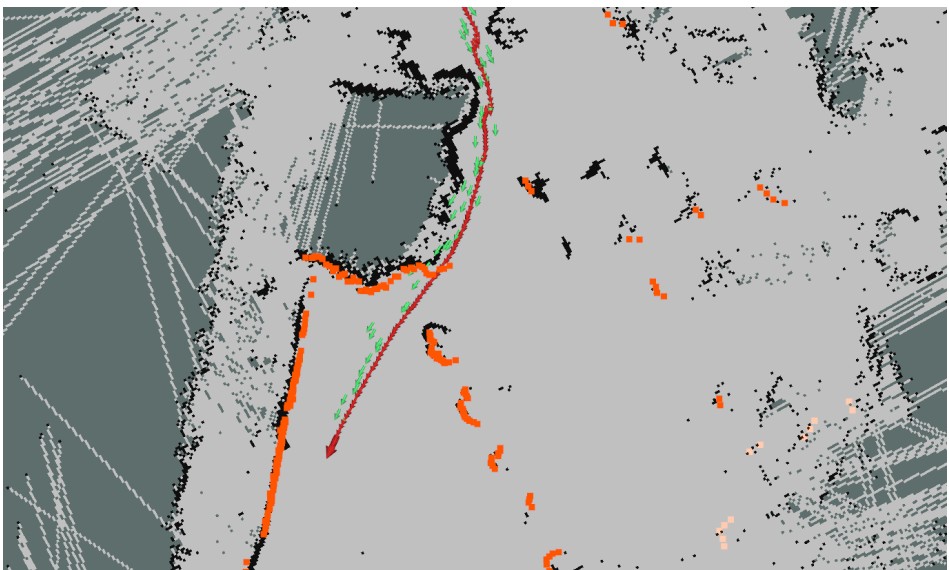

**Figure 9.** Example of localization using GNSS and LiDAR data.

Figure 10 compares the proposed method with the localization sources that it combines (GNSS and original AMCL), plotting the error along a recorded sequence. The method that combines LiDAR and GNSS information results in a more stable localization, avoiding peaks on localization error or the kidnapped robot problem shown at the end of the sequence by AMCL algorithm, while improving GNSS accuracy (0.25 m of mean error compared to 0.38 m for GNSS localization) due to the use of the LiDAR and map information.

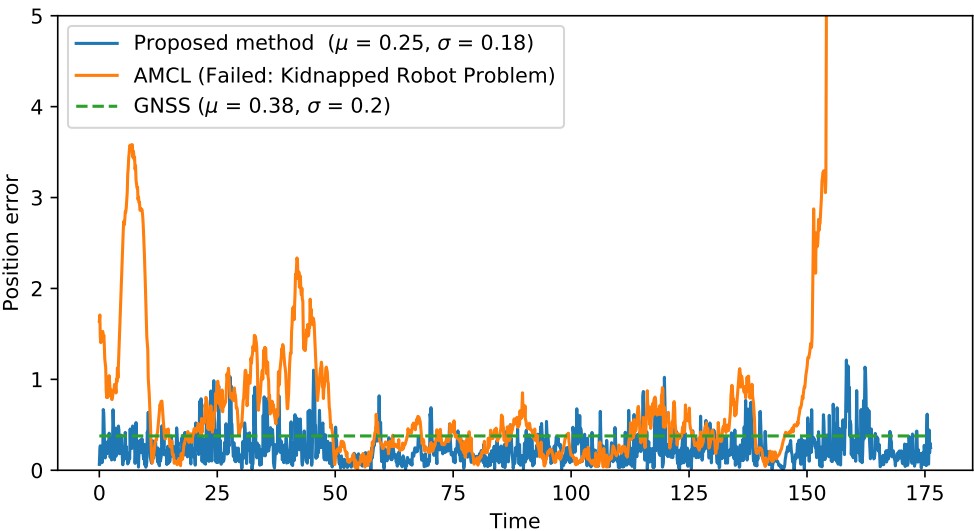

**Figure 10.** Comparison of the localization accuracy of the proposed method, original adaptive Monte Carlo localization (AMCL) and GNSS.

*4.2. Environment Simulation*

Testing the previous architecture software modules and algorithms under real circumstances is of critical importance for validating the proposed solutions. However, to test the different features inside the proposed platform, safety and legal concerns must be addressed, which is time-consuming and limits the authenticity of the results obtained, as the vehicle cannot be tested under real traffic and driving conditions.

To aid in this testing issue, the previously described simulation module (Section 3.10) is used. This module is not only capable of simulating the proposed platform inside the pre-designed CARLA environments but is also able to recreate real road networks in order to test how these proposed algorithms perform under real driving conditions without the need for physical testing in the environments and the road networks.

For this purpose, OpenStreetMap data are used to recreate any real road network desired inside the CARLA simulator. To achieve this, the raw OpenStreetMap data are first preprocessed using the osmfilter to remove all the unnecessary features of the environment (pedestrian streets, railroads, rivers, buildings, etc.) in order to make the data lighter and easier to handle. Afterwards, the filtered data are transformed into an OpenDrive format using SUMO [25] (Simulation of Urban Mobility), which performs an automatic conversion. Once the road network is in OpenDrive format, a CARLA world can be directly generated from it using the OpenDrive standalone feature of the simulator. Moreover,OpenStreetMap data are geo-referenced; thus, the GNSS coordinates are preserved during the CARLA world generation, meaning that the artificial GNSS data generated by the simulator match the real readings that can be obtained in the real environment.

Furthermore, OpenStreetMap data can be used to obtain essential knowledge of the environment in order to complete the simulated road network with additional data. Using the proposed OpenStreetMap feature described in the mapping software module (Section 3.4), information such as the number of lanes, the road direction, the location of pedestrian crossings, the names of the streets or the maximum speed permitted can be obtained and added to the simulated environment.

Additionally, agents such as pedestrians, cyclists and other vehicles can be added to the digital environment, which is able to simulate standard traffic and driving conditions (Figure 11).

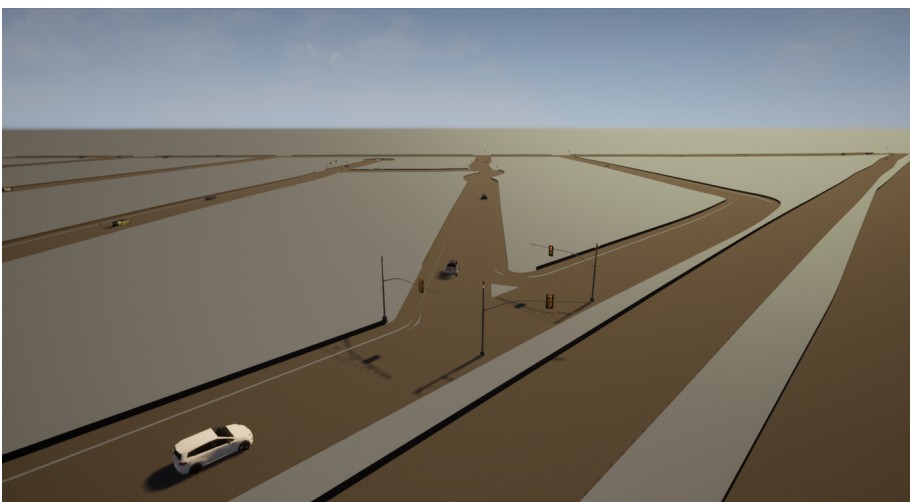

**Figure 11.** CARLA world generated from an OpenDrive file, with agents.

Consequently, the simulated environment represents an exact copy of the real road network that includes all the static data that the real environment can offer. Therefore, the results that can be obtained in the experiments fulfilled in the simulation can be reliably extrapolated to the real environment that it mimics.

Inside this simulated environment, the proposed platform can be simulated as in any other CARLA world, as described in Section 3.10. In this way, localization, navigation, mapping, perception and safety modules can be tested and interchanged between the different variants, allowing fast testing and analysis, even for experimental or newly developed features that do not need to pass a minimum safety revision to be tested, unlike in the real platform.

In order to validate this environment recreation feature, a part of Ávila's road network—a Spanish city—has been replicated inside CARLA Simulator using the previously described methodology. Even though the quantity of OpenStreetMap data available for Ávila is limited, the road network has been recreated successfully both inside CARLA and ROS, creating a fully drivable digital road network of a part of the city in CARLA and obtaining all the environment relevant additional information from OpenStreetMap and publishing it through ROS, making it available for all the systems.

In this scenario, the three platforms—LiDAR, the GNSS and the IMU—have been used to perceive both the environment and the intrinsic state of the vehicle. Then, different variants of the software modules have been successfully tested, such as the unscented Kalman filter, regarding the localization, or the Stanley controller for navigation and control (see Figure 12). In consequence, we determined that the digital platform inside the simulated environment works analogously to the real platform, implying that tests can be performed inside the simulator to obtain preliminary results or useful indicators regarding the performance of the different modules or the performance of the platform as a whole.

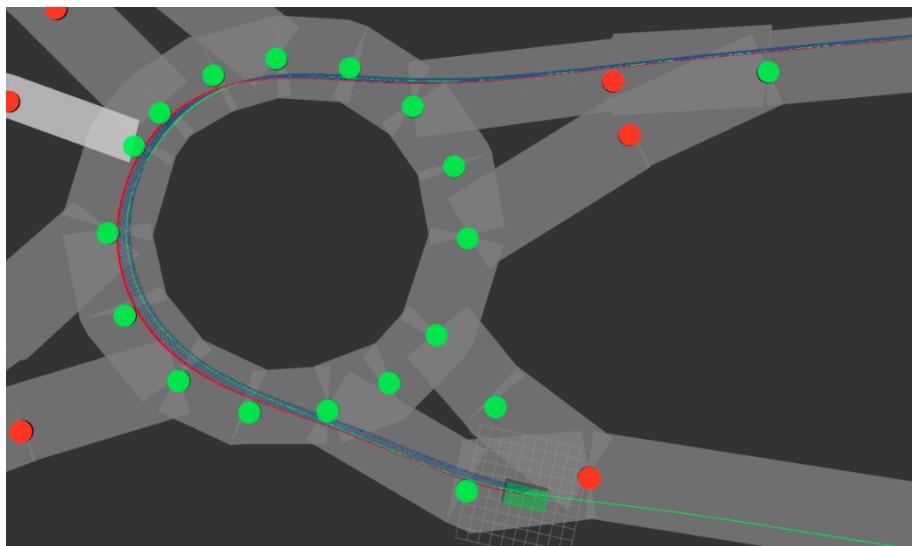

**Figure 12.** Results of testing the simulated Autonomous Testing Platform for Insurance Research (ATLAS) inside a digital copy of Avila's road network. In green: the path to follow, in blue: the GNSS odometry, in red: the unscented Kalman filter (UKF) localization output.

### 4.3. Human Factors

Although most of the research on autonomous vehicles has usually focused on navigation and perception problems, the interaction between the car and other vulnerable road users (VRUs) has recently attracted increased attention. These VRUs include pedestrians or cyclists, and the interactions between these and the vehicles are of special importance in urban areas. Previous works have tackled how pedestrians react when facing a driverless vehicle, as well as how to identify pedestrians' intentions. For instance, in [26], a human–machine Interface (HMI) was presented. The behavior of pedestrians was analyzed, concluding that a communication system proves to be positive in the interaction between the pedestrian and the vehicle. Moreover, the behavior of pedestrians was further studied in [27], where the vehicle detected the pose of pedestrians in order to predict their intentions.

With the purpose of continuing this research, the proposed platform represents an ideal asset to perform this kind of experiments. The versatility of the roof rack allows us to quickly adapt the hardware to the experimental requirements. For example, in [23], two different HMIs are installed in order to enhance the communications between the vehicle and the pedestrians and analyze the behavior of VRUs when approaching the vehicle. The experimental setup, which is presented in Figure 13, consists of two different HMIs: first, a large screen is used to inform the pedestrian about the intention of the vehicle; alternatively, a pedestrian traffic light is also installed to indicate to the pedestrian whether it is safe to cross or not.

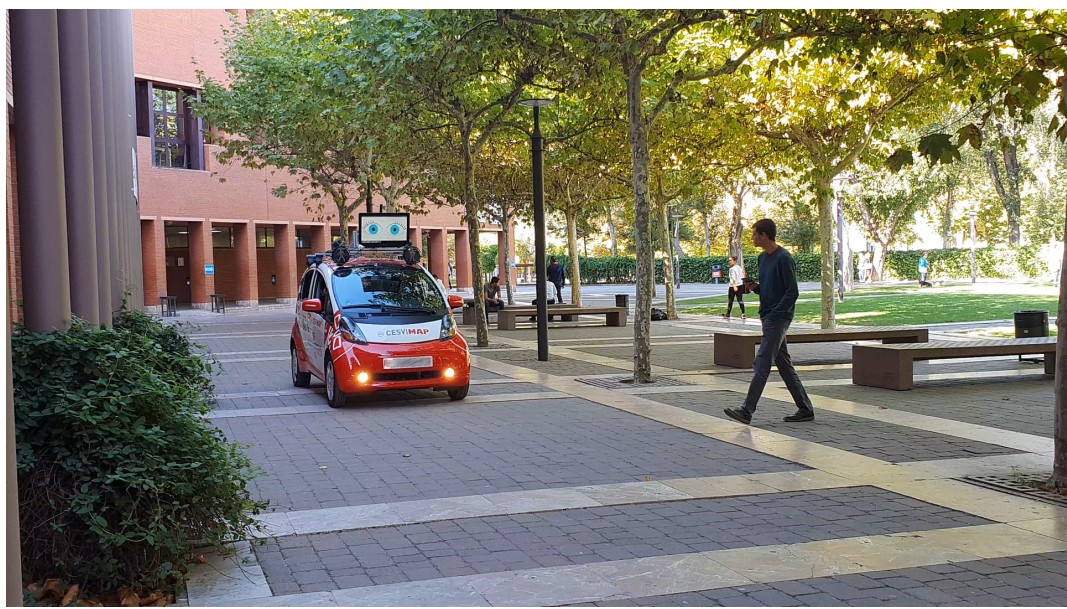

**Figure 13.** Experimenting with different human–machine interfaces (HMIs).

### 4.4. V2X Research

The proposed platform has been also used to extend the experiments presented in [21]. In this paper, a V2X communications architecture is proposed for off-road autonomous vehicles; however, all experiments are carried out in golf cart-like vehicles, with a limited maximum speed. For this reason, several additional experiments have been carried out using the proposed platform in order to further validate the V2X architecture in an actual car moving at medium and high speeds.

As described in Section 3.9, the vehicle has Internet connection capabilities through a 4G router. This connection is used to connect to a VPN, as introduced in [21], which is only accessible to specific permitted agents (other vehicles, infrastructure or pedestrians). This VPN grants an extra security layer to perform the communication, regardless of the distance between the agents, which makes it a viable option for communicating at high speeds.

In order to validate the proposed V2X architecture in this system, two new scenarios have been created. In each scenario, the presented vehicle is moving while communicating with a static golf cart. The information shared between each vehicle consists of position, orientation and vehicle status, and it is transmitted at 20 Hz. The new experimental scenarios are detailed below:

- **Highway:** The vehicle is moving on a highway at an average speed of 100 Km/h;
- **Urban:** The vehicle is moving in urban and peri-urban areas at an average speed of 45 Km/h.

Following the original metrics used in [21], Figure 14 shows the round-trip time (RTT) of the network packets that are exchanged between vehicles in each scenario.

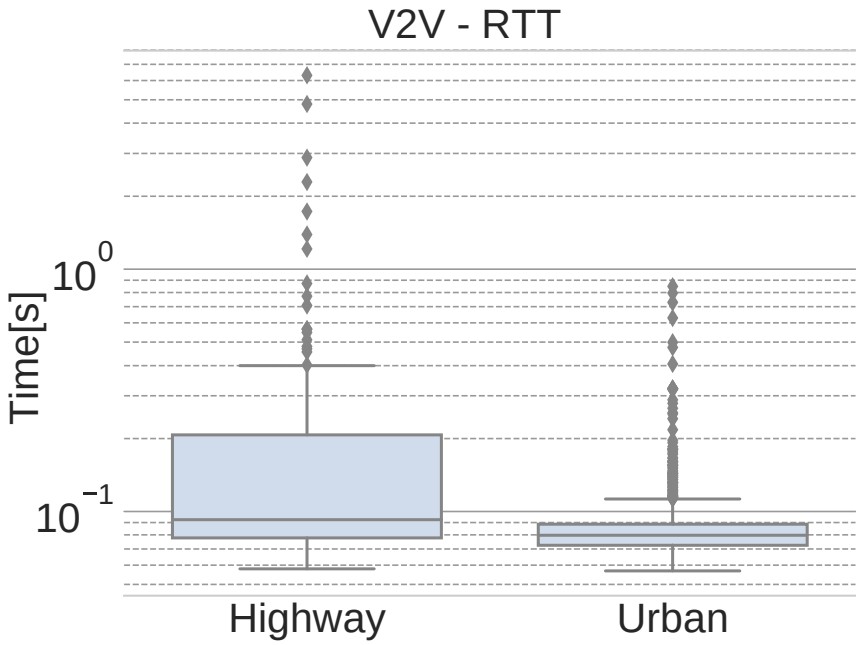

**Figure 14.** Network packet round-trip time (RTT) from vehicle to vehicle (V2V) experiments.

As expected, the number of outliers increases when moving at higher speeds. Due to the use of 4G, the handover between access points may also influence this high number of outliers. Nevertheless, the average RTT values, presented in Table 1, are below 100 ms, which is adequate for non critical messages.

**Table 1.** Statistics of V2V RTT values. IQR: interquartile range.

| Mean (s) | | IQR (s) | | Max (s) | |
|---|---|---|---|---|---|
| Urban | Highway | Urban | Highway | Urban | Highway |
| 0.074 | 0.076 | 0.062 | 0.07 | 0.8 | 6.3 |

In addition to the average RTT values, Table 1 also shows the interquartile range (IQR) of the data and the maximum RTT values obtained for each scenario. The high maximum values, along with the rest of the outliers, are mainly due to lost packets that must be re-transmitted, thus producing a higher RTT value. This problem does not affect continuous, non-critical flows of information, such as localization or images, because those streams do not require re-transmission when a packet is lost. Instead, the next (and newest) message is sent.

*4.5. Demonstrations and Competitions*

Due to its great versatility, the presented vehicle is also a great platform for demonstrations and competitions. For example, the perception algorithms and the automation capabilities were showcased in a demonstration event for autonomous vehicles during the IROS 2018 conference, as shown in Figure 15.

Furthermore, the team at the Intelligent Systems Lab participated in the Dubai World Challenge for Self-Driving Transport 2019 with the proposed platform, successfully classifying to the finals and winning the second-place award.

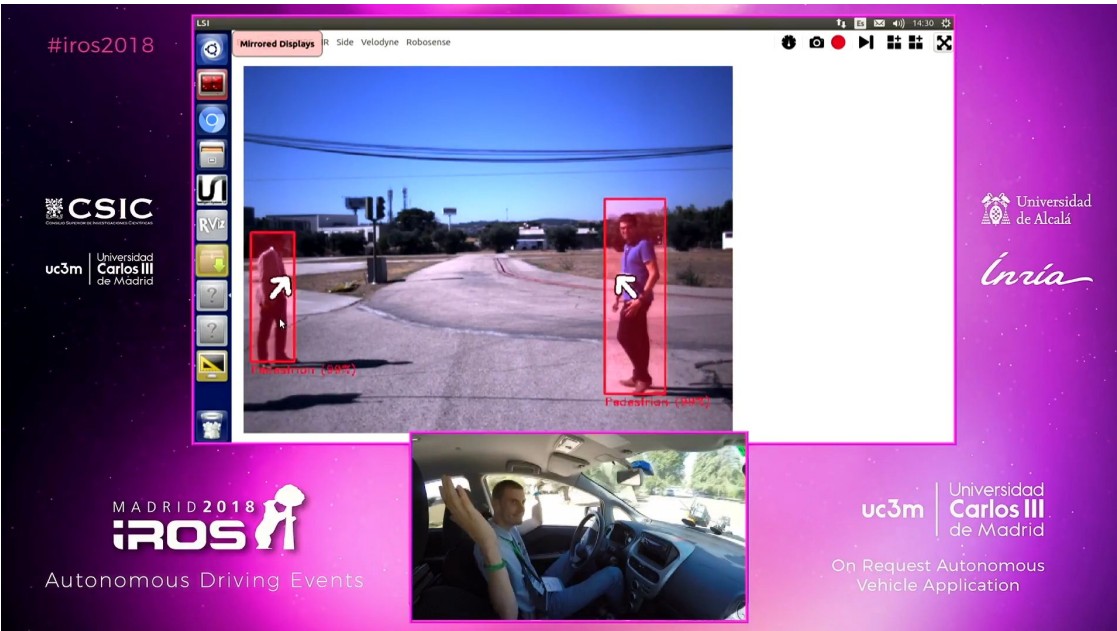

**Figure 15.** IROS18 autonomous driving event.

## 5. Conclusions and Future Work

In this article, a new research platform designed jointly by MAPFRE-CESVIMAP, LSI-UC3M and INSIA-Universidad Politécnica was presented. The vehicle has been designed to evaluate autonomous vehicle technologies and their impact in the field of automobile insurance in the coming years. To this end, an ad-hoc architecture designed for research into high-level systems for autonomous vehicles, both for control and perception, has been presented. The advances presented by this platform include perception, communications, human factors and self-awareness technologies, all of which are at the forefront of current advances in autonomous and connected vehicles. In addition to presenting the platform, different use cases have been presented in which the use of this innovative platform has allowed us to propose new results advancing the state of the art. These examples include demonstrations at international conferences such as IROS 2018, advanced perception algorithms and location and tracking algorithms, as well as communications systems, in addition to international competitions, such as the second place achieved in the Dubai World Challenge for Self-Driving Transport 2019. These use cases have demonstrated the viability of the system and the usefulness of advancing and testing autonomous driving algorithms.

The future work on the development of this platform will seek to demonstrate the suitability of new location algorithms in urban environments, based on vision and LiDAR technologies, as well as the fusion of this information with GNSS systems. On the other hand, we aim to improve the detection systems by means of LiDAR and camera fusion based on modern sensorial fusion technologies. Finally, we will evaluate the capacity of these systems in real driving environments and investigate new techniques for communication with pedestrians.

**Author Contributions:** Conceptualization, P.M.-P., M.Á.d.M., R.E.-M. and F.M.M.; methodology, P.M.-P., M.Á.d.M. and F.M.M.; software and validation, M.P., M.Á.d.M. and F.M.M.; resources, A.A.-K, D.M., R.E.-M. and F.G.; writing original draft preparation, F.M.M. and M.Á.d.M.; writing review and editing, F.M.M. and F.G.; supervision, A.A.-K. and F.G.; funding acquisition, A.A.-K., D.M., R.E.-M. and F.G. All authors have read and agreed to the published version of the manuscript.

**Funding:** Research was supported by the Spanish Government through the CICYT projects (TRA2016-78886-C3-1-R and RTI2018-096036-B-C21) and the Comunidad de Madrid through SEGVAUTO-4.0-CM (P2018/EMT-4362) and PEAVAUTO-CM-UC3M.

**Conflicts of Interest:** The authors declare no conflict of interest.

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
