# Peer review of "A Research Platform for Autonomous Vehicles Technologies Research in the Insurance Sector"

_applsci, doi:10.3390/app10165655_

Round 1

Reviewer 1 Report

This paper presents a well-established autonomous vehicle. The whole structure is very nice. It provides a detailed description about how to construct an autonomous vehicle.

Herein, I have some small questions which can help further improve this paper.

  1. The whole paper seems like a project report, not an academic article. I recommend that the authors should focus on more recent technologies which are related to their autonomous vehicle. For example in the motion and path planning field,

[1] Wang, Jiankun, Max Q-H. Meng, and Oussama Khatib. "EB-RRT: Optimal Motion Planning for Mobile Robots." IEEE Transactions on Automation Science and Engineering (2020).

[2] Wang, Jiankun, et al. "Neural RRT*: Learning-Based Optimal Path Planning." IEEE Transactions on Automation Science and Engineering (2020).

  1. Fig.2 seems not good enough. I mean that this figure can be improved by adding a small picture of Lidar, IMU to the corresponding part. This version is very simple and should be refined.
  2. Since there are many kinds of autonomous vehicles all over the world, it would be great if the authors can provide a brief introduction or comparison with others.

Author Response

This paper presents a well-established autonomous vehicle. The whole structure is very nice. It provides a detailed description about how to construct an autonomous vehicle.

Authors’ response:We would like to thank the reviewer for the comments and suggestions that have allowed us to improve the quality of the article as well as to correct the errors of the first version. We hope that this new version will meet his expectations.

Herein, I have some small questions which can help further improve this paper.

The whole paper seems like a project report, not an academic article. I recommend that the authors should focus on more recent technologies which are related to their autonomous vehicle. For example in the motion and path planning field,

[1] Wang, Jiankun, Max Q-H. Meng, and Oussama Khatib. "EB-RRT: Optimal Motion Planning for Mobile Robots." IEEE Transactions on Automation Science and Engineering (2020).

[2] Wang, Jiankun, et al. "Neural RRT*: Learning-Based Optimal Path Planning." IEEE Transactions on Automation Science and Engineering (2020).

Authors’ response:We appreciate the help of the reviewer to provide feedback of the paper, specifically for this comment, in this new version we tried to highlight the most recent technologies, adding some new references including the ones recommended. 

Fig.2 seems not good enough. I mean that this figure can be improved by adding a small picture of Lidar, IMU to the corresponding part. This version is very simple and should be refined.

Author’s response: We appreciate the comment of the reviewer regarding to this specific point. We agree that the figure on its current format is a bit simple. However we strongly believe that the simplification of the figure provides generalization. The architecture presented has been implemented for ATLAS platform, however is being tested in different platforms on which the Intelligent System Lab works, including autonomous buses and golf carts. The addition of images of specific sensors would lead to the misconception that the architecture was designed specifically for those sensors.

Since there are many kinds of autonomous vehicles all over the world, it would be great if the authors can provide a brief introduction or comparison with others.

Author’s response: We understand the concern of the reviewer regarding to this point. In order to clarify it, we have included in the introduction section some references to the most well known platforms .

Reviewer 2 Report

The paper presents the overall system of a autonomous vehicle. There is no algorithm details in the paper. The authors just introduce their techniques and show the results. Based on the application results, it is credible of the realization of autonomous driving. The paper is suitable to be accepted in current form.

A minor issue: Figure 1 is not referred in the text.

Author Response

The paper presents the overall system of a autonomous vehicle. There is no algorithm details in the paper. The authors just introduce their techniques and show the results. Based on the application results, it is credible of the realization of autonomous driving. The paper is suitable to be accepted in current form.

A minor issue: Figure 1 is not referred in the text

Authors’ response:We would like to thank the reviewer for the comments and suggestions that have allowed us to improve the quality of the article as well as to correct the errors of the first version. We hope that this new version will meet his expectations. Regarding to the figure 1 issue, it has been corrected accordingly.

Reviewer 3 Report

The design which presented in this paper does not match with the safety standards as example ISO 26262. Why do you use 3 LiDAR not 4? Why do not use radar? 

What happen for this car with different environmental situations? 

How do you fuse the Lidar with GPS and IMU?

Why did you mention GPS not GNSS?

Which type of RTK do you use in your staudy?

.

.

.

Author Response

The design which presented in this paper does not match with the safety standards as example ISO 26262. Why do you use 3 LiDAR not 4? Why do not use radar? 

Authors’ response:We would like to thank the reviewer for the comments and suggestions that have allowed us to improve the quality of the article as well as to correct the errors of the first version. We hope that this new version will meet his expectations. Regarding to  use of ISO 26262, we have to remind the reviewer that this platform is a research platform, and subsequently it is designed for testing and development. On the other hand, ISO 26262 “is an international standard for functional safety of electrical and/or electronic systems in production automobiles defined by the International Organization for Standardization (ISO)” Thus as the vehicle is not a production vehicle, it is not expected to follow this standard. However we can guarantee that any approach tested and developed on this car which may go to production, is developed under this ISO standard. 

The selection of 3 Lidars guarantees a full 360 coverage, the location and orientation of them was selected based on a meticulous simulation. The addition of more units would require more processing requirements while not increase the field of view. In order to clarify this aspect we included this explanation in the text in section 2.2. 

What happen for this car with different environmental situations? 

Authors’ response:We  understand the concerns of the reviewer regarding to this aspect. Particularly complicated environments is a current trend in the research community, and as such is being investigated within the framework of this cooperation, current results are promising, however not yet ready to be published. We have to remind that this is a research platform. In order to clarify this aspect we included a sentence explaining this situation in the text in section 3.5.

How do you fuse the Lidar with GPS and IMU?

Authors’ response: As stated in the text, the laboratory has developed different and variated fusion algorithms, including H infinity ( section 3.3)  and particle filter (section 4.1) However in many of the modern devices IMU and GPS are provided together in the output, as in the PixHawk device. 

Why did you mention GPS not GNSS?

Authors’ response: we appreciate the help of the reviewer to correct this mistake, in this new version any GPS mention was changed to GNSS.

Which type of RTK do you use in your staudy?

Authors’ response: we appreciate the help of the reviewer to correct this lack of information, in this new version we have specified the RTK system used.

Reviewer 4 Report

The paper is well-structured so that it can be published as it is.

Author Response

The paper is well-structured so that it can be published as it

Authors’ response:We would like to thank the reviewer for the support.

Round 2

Reviewer 3 Report

Ok.